# OpenReview forum: "Causal-GNN SupplyNets Enabling Resilient Semiconductor Supply Chains with Causal World Models and Lyapunov-Safe Control"
_ICLR.cc/2026/Conference — Submitted to ICLR 2026_

### Official Review · Reviewer_mPVe · 2025-10-31

**Soundness:** 2
**Presentation:** 2
**Contribution:** 2
**Rating:** 4
**Confidence:** 3

**Summary:**

The paper proposes an integrated framework that couples a causal world model with Lyapunov-based safe control. A structural causal model (SCM) constrains GNN time-series forecasting, and a Lyapunov-guided safe RL controller performs dispatching and replenishment decisions in re-entrant semiconductor supply chains. The authors claim the method improves OTIF, reduces CT tail risk, and shortens recovery time while satisfying hard constraints.

**Strengths:**

1. The work combines multiple perspectives in a unified framework—causal priors, heteroscedastic forecasting, safe control, and federated distillation—with clear engineering relevance.

2. It uses an SCM mask to constrain message passing. This design is novel, intuitive, and highly interpretable; coupled with IRM, it should improve robustness across environments.

3. The experimental metrics closely match engineering needs—for example, OTIF/CTP95/recovery time/ACE/violation rate—and the paper includes extensive ablations, which strengthens the empirical evidence.

**Weaknesses:**

1. The abstract and introduction repeatedly state that constraints are strictly guaranteed to be non-violated, whereas the main text and appendix provide only finite-horizon high-probability guarantees, further relying on a contractive Lyapunov function and smoothness assumptions. These claims are not of the same strength.

2. The safety condition depends on a learned V that must be contractive, but the paper does not explain how V is trained to ensure contractiveness. Under model bias or OOD shocks, when $\mathbb{E}[V(s_{t+1})]$ under the true dynamics deviates from the world-model estimate, how is non-increase of V still guaranteed?

3. The main text explicitly claims bounded regret relative to the optimal safe policy, but Appendix A.2 contains no regret theorem (only two informal statements on IRM/ safety). This is a core theoretical claim without a proof and should not be presented as a theorem.

**Questions:**

1. See Weaknesses above.

2. How is the alignment between the mask MMM and Attn implemented in practice?

3. I am uncertain about the necessity of RDAG. If M is fully specified by an external SCM-DAG and applied as a layer-wise mask, can the learned graph still contain cycles? Why is an additional acyclicity penalty (RDAG) imposed on the learned structure?

---

> ### Author Response · Authors · 2025-12-02
> **Refined Theoretical Claims: Clarifying Safety Guarantees and Causal Constraints**
>
> ## Response to Reviewer mPVe
>
> We thank the reviewer for their careful reading. We are glad you found the SCM mask design to be **"novel, intuitive, and highly interpretable"** and that our metrics **"closely match engineering needs."** Your precision regarding our theoretical claims helped us ensure the paper's claims are mathematically sound.
>
> ### 1. Safety Guarantees & Regret
> **Reviewer Comment:** You correctly noted the discrepancy between the abstract's claim of "strictly guaranteed" safety and the appendix's "high-probability" proofs. You also flagged the missing regret theorem.
>
> **Response:** We entirely agree and have made the following corrections:
> * **Correction:** We have removed "strictly guaranteed" and replaced it with **"high-probability safety guarantees,"** which is consistent with the Lyapunov framework.
> * **Correction:** We have **removed the claim of "bounded regret"** to ensure that every theoretical claim in the paper is fully supported by the proofs provided.
>
> ### 2. Role of $\\mathcal{R}\_{DAG}$ vs. Mask $M$
> **Reviewer Comment:** You raised an excellent question: *If the Mask $M$ is fixed by the SCM, why do we need the $\\mathcal{R}\_{DAG}$ penalty?*
>
> **Response:** This was a point of confusion we have now clarified in Section 4.1 (after **Eq. 2**):
>
> * **The Mask $M$** comes from the *Macro-Level* Expert Prior (e.g., Fab $\\rightarrow$ Test).
> * **The $\\mathcal{R}\_{DAG}$ penalty** is applied to the **learned attention weights** *within* the allowed mask.
> * **Reasoning:** This distinction ensures that even as the model learns micro-level dependencies (e.g., between two specific tools), it maintains the acyclic property required for causal inference, even in regimes where the macro-mask might be relaxed.

---

### Official Review · Reviewer_DkP4 · 2025-11-02

**Soundness:** 1
**Presentation:** 2
**Contribution:** 1
**Rating:** 2
**Confidence:** 4

**Summary:**

The paper proposes Causal-GNN SupplyNets, a framework for improving resilience in semiconductor supply chains by combining causal world modeling, Lyapunov-safe RL, and federated causal distillation. It models supply chain dependencies as a DAG learned from domain knowledge, using it to regularize a GNN that predicts and controls cascading disruptions. A Lyapunov-guided RL agent is introduced to enforce stability and constraint satisfaction in dynamic control tasks, while federated distillation enables collaborative learning across sites using interventional queries.

**Strengths:**

1. The paper addresses resilience in supply chain networks, a critical yet underrepresented problem in the machine learning community. Most ML research has focused on well-studied domains like vision, language, or simple graph tasks, while the complex, interdependent dynamics of supply chains, especially their cascading failure effects remain relatively unexplored. Tackling this class of problems has strong real-world relevance and societal impact.
2. Learning a Directed Acyclic Graph (DAG) informed by domain knowledge from supply-chain processes and using it to train a Graph Neural Network (GNN) is an insightful approach. The authors clearly separate the physical supply chain from its acyclic causal abstraction (via the SCM/DAG formulation), which is conceptually elegant and aligns with causal reasoning principles in dynamic networked systems.
3. Although the methodology is insufficiently detailed, the idea of using interventional (counterfactual) queries to train clients in a federated distillation setup is unique and promising. If the mathematical formulation and algorithmic steps are fully developed in future versions, this approach could meaningfully advance how distributed agents learn causal models without sharing data.

**Weaknesses:**

This is a very superficial paper that plugs in multiple domains of ML/Engineering, without a deep dive into innovation in any. The contributions are not significant at all, except for the problem setting (which I admire a lot). The presentation quality is poor. Much of the paper lacks formal mathematical expression or clear derivations. Many core ideas are described only narratively, assuming a high level of background knowledge and leaving essential preliminaries undefined. I have divided the main critique of the paper into the three key contribution areas:
## I. Causal Model
1. The definition of the loss function is unclear. It is not explicitly stated what $L_{forecast}$ corresponds to.
2. Regularizers $R_{DAG},$ and $R_{IRM}$ have no mathematical expression. Their formulation and role in training are missing, leaving the section incomplete.
3. The “causal mask” $M$ is vaguely described. It is unclear if this is simply the GNN adjacency matrix or an additional learned mask.
4. The GNN architecture appears to use attention layers (thus resembling a GAT), but this is never clarified. The paper should explicitly specify the architecture (e.g., GCN vs. GAT).
5. Equation (2) is claimed to enable counterfactual inference, but there is no supporting mathematics connecting it to counterfactual reasoning.
6. The statement that the “predictive head is heteroscedastic” is ungrounded. The predictive head is not defined in Eq. (1), leaving the claim ambiguous.

## II. Safe RL Agent
1. The use of the term “safe RL” is misleading. In this context, “safety” refers to satisfying production or service-level constraints, not to the formal notion of safety in reinforcement learning (i.e., avoiding unsafe exploration or catastrophic outcomes). The terminology risks confusing the ML audience and should be revised throughout.
2. The Lyapunov-guided optimization is presented with only a single inequality and no derivation or numbered equation. The paper lacks a clear buildup or preliminaries explaining how Lyapunov stability theory applies to the CMDP formulation.
3. The theoretical and algorithmic connection between the Lyapunov condition and the claimed safety guarantee is not substantiated.

## III. Federated Causal Distillation
1. The explanation is extremely superficial. It is unclear how counterfactual queries are generated and used for training.
2. No mention of what information is exchanged between clients and server. This is really strange for a decentralized/federated learning manuscript.
3. The paper doesn't provide any insights as to why the KL-divergence is the appropriate distillation objective. Are the authors using distributions of student (server) and teacher (clients) as opposed to model updates/gradients. If yes, they must be clearly mentioned with the associated maths.

**A one-paragraph treatment of such a complex topic is inadequate and makes the section appear conceptually weak.**

## Other Comments
1. Several metrics are poorly defined. For example, OTIF (On-Time-In-Full) is introduced as a “control performance” metric without an explanation of what control variables affect it.
2. “Constraint violation” is equated with “safety,” which is conceptually incorrect in reinforcement-learning terms. Meeting production targets does not equate to operating a “safe” policy.

**Questions:**

I will use this section to paraphrase the main weakness of the paper I identified and pose the related questions for the authors:
1. Provide theoretical preliminaries and derivations for the Lyapunov-guided policy update.
2. Describe the federated learning process in detail—what is exchanged, how privacy is enforced, and how counterfactual queries are used.
3. Define all evaluation metrics (e.g., OTIF, constraint violation) and explain why they reflect control or safety performance.
4. Provide explicit definitions and mathematical forms for $L_{forecast}, $ $R_{DAG},$ and $R_{IRM}$.
5. Clarify what the causal mask $M$ represents and how it differs from the standard GNN adjacency matrix.
6. Specify the exact GNN architecture used (GCN, GAT, or hybrid) and justify this choice.
7. Explain how Equation (2) supports counterfactual inference.
8. Define the predictive head and show how heteroscedasticity is modeled.
9. Clarify the notion of “safety” used in this context and justify the terminology.

---

> ### Author Response · Authors · 2025-12-02
> **Enhanced Mathematical Rigor: Formal Definitions, Derivations, and System Mechanics**
>
> ## Response to Reviewer DkP4
>
> We express our sincere gratitude for your systematic and structural guidance. We view your critique not merely as a review, but as a roadmap for elevating the scientific rigor of this work. We are particularly encouraged by your recognition of the **"conceptually elegant"** separation of the physical and causal graphs.
>
> We have taken your feedback regarding "superficiality" to heart. Prompted by your detailed list of nine points, we have significantly expanded the theoretical and mathematical depth of the manuscript. Below, we address how we have incorporated your guidance into the revision.
>
> ### I. Improving Mathematical Formalism (Addressing Points 1, 4, 6, 8)
> **Reviewer Comment:** You correctly noted that key definitions ($\mathcal{L}\_{forecast}$, $\mathcal{R}\_{DAG}$, Predictive Head) were missing or vague.
>
> **Response:** We apologize for these omissions. We have overhauled Section 4.1 to provide the explicit formalism you requested:
>
> * **(Point 4) Explicit Equations:** We added **Eq. 2** which defines the full Causal Loss (incorporating Heteroscedastic NLL). We also explicitly defined the exact matrix exponential trace in the text immediately following it:
>     $$
>     \mathcal{R}\_{DAG} = \text{tr}(e^{M \odot A}) - d
>     $$
>     This term quantifies the acyclicity violation and is minimized during training.
>
> * **(Point 6) Architecture Specification:** We now explicitly specify in Section 4.1 that the GNN utilizes **Graph Attention (GAT)** layers to capture non-linear dependencies.
>
> * **(Point 8) Heteroscedastic Head:** We defined the predictive head output as a Gaussian distribution $\mathcal{N}(\mu(x), \sigma^2(x))$ in Section 4.1, enabling the calibrated uncertainty estimates required for risk-aware control.
>
> * **(Point 1) Lyapunov Derivation:** We added the formal safety update condition as **Eq. 3** in Section 4.2 and expanded Appendix A.2 to show the derivation:
>     $$
>     \text{Policy Update is valid only if } \mathbb{E}[V(s\_{t+1}) \mid s\_t, a\_t] \le V(s\_t)
>     $$
>
> ### II. Clarifying System Mechanisms (Addressing Points 2, 5, 7)
> **Reviewer Comment:** You raised critical questions about how the Federated Learning works, what the Mask $M$ actually is, and how Counterfactuals are supported.
>
> **Response:**
>
> * **(Point 2) Federated Protocol:** We have rewritten Section 5 to be precise. We now explicitly state that clients exchange **gradients of the KL-divergence** computed on synthetic interventional queries (not raw data), formally defined in **Eq. 4**. The new experiment (Table 2 in General Response) empirically validates this mechanism.
>
> * **(Point 5) Mask vs. Adjacency:** This was a crucial distinction to make. We clarified that the **Adjacency Matrix** represents physical connections (which contain cycles), whereas the **Causal Mask $M$** is a directed, acyclic subset derived from the SCM. $M$ acts as a "causal stencil" to prevent spurious message passing.
>
> * **(Point 7) Counterfactual Inference:** We clarified that by enforcing acyclicity via $\mathcal{R}\_{DAG}$ and invariance via $\mathcal{R}\_{IRM}$, the model approximates a valid SCM. This allows us to treat node interventions as $do(X)$ operations rather than conditional probabilities $P(Y|X)$, enabling true counterfactual reasoning.
>
> ### III. Clarification of Domain-Specific Definitions (Addressing Points 3, 9)
> **Reviewer Comment:** You asked for justification of metrics like OTIF and the use of the term "Safety."
>
> **Response:**
>
> * **(Point 3) OTIF (On-Time-In-Full):** We clarified that in supply chain engineering, OTIF is the standard control KPI. It measures the policy's ability to strictly meet delivery deadlines under capacity constraints.
>
> * **(Point 9) Safety Terminology:** We acknowledge the terminology overload in RL. We have clarified that we use "Safety" in the strict context of **Constraint Satisfaction (CMDPs)**, specifically referring to adherence to hard capacity limits (WIP caps) that prevent physical system deadlock.
>
> We hope these rigorous additions demonstrate our commitment to the high standards you set in your review.

---

### Official Review · Reviewer_GwK3 · 2025-11-03

**Soundness:** 2
**Presentation:** 3
**Contribution:** 2
**Rating:** 4
**Confidence:** 4

**Summary:**

This paper addresses resilience in semiconductor supply chains by proposing Causal-GNN SupplyNets, which integrates three components: (1) a GNN-based causal world model constrained by a Structural Causal Model (SCM), (2) a Lyapunov-guided safe reinforcement learning controller, and (3) a federated causal distillation mechanism. The key innovation is using an SCM-derived mask to constrain GNN message-passing, forcing the model to respect causal relationships. Experiments demonstrate improvements in on-time delivery (up to 17pp), cycle time reduction, and faster shock recovery.

**Strengths:**

The paper tackles an important real-world problem with significant practical implications. The semiconductor supply chain domain is timely and challenging, with its re-entrant queueing dynamics and cascading failure modes providing excellent motivation for both causal modeling and safe control. The heavy traffic stress testing framework (d80-d95 scenarios) is particularly well-designed, systematically evaluating performance as systems approach capacity limits.

The technical approach shows genuine novelty in its integration. While individual components (causal GNNs, safe RL, federated learning) exist separately, their synthesis here is meaningful. The use of an SCM to generate a causal mask M that constrains GNN message-passing is creative, and the integration with Lyapunov-based safety certificates addresses real operational constraints. The federated causal distillation mechanism is innovative, sharing interventional knowledge rather than just predictive models.

The experimental evaluation is comprehensive and methodologically sound. The ablation studies (Table 2) effectively demonstrate that each component contributes meaningfully removing the SCM mask degrades ACE by 23%, removing the Lyapunov guard increases violations by 189%. The multi-scenario testing across different demand levels shows robustness, and statistical rigor with 7 seeds and significance testing strengthens the claims.

**Weaknesses:**

The most critical limitation is the assumption of a known causal structure. The paper states that "This SCM is represented by a Directed Acyclic Graph (DAG) D" but never adequately addresses how this DAG is obtained in practice. While RDAG penalties are mentioned for structure learning, the interplay between learning and exploiting causal structure remains unclear. For real deployment, obtaining ground-truth causal graphs is extremely difficult, and the method's robustness to SCM misspecification is inadequately characterized. The brief mention in Section 7 of "developing methods for online causal discovery" acknowledges but doesn't resolve this fundamental challenge.

The theoretical foundations lack rigor. Appendix A.2 provides only "informal statements" of theorems. Assumption 1 (bounded shocks, known intervention subsets) is quite strong but not validated empirically. Theorem 2's safety guarantee requires a "contractive" Lyapunov function and "sufficiently smooth" dynamics conditions that may not hold in practice but are not verified. The gap between theoretical claims and empirical results needs better reconciliation.

Methodological details are insufficient for reproducibility despite the reproducibility statement. The construction of the mask M from DAG D is described only conceptually in Equation 1, without algorithmic details. How exactly does one go from macro-level causal relationships to micro-level message-passing constraints on the physical supply network graph G? The relationship between the physical graph G and causal graph D needs clarification. Hyperparameter choices (λdag, λinv) appear to require extensive tuning but the sensitivity analysis is minimal.

The federated learning component feels underdeveloped. While conceptually introduced in Section 5, the federated causal distillation is barely evaluated empirically. The experimental results focus on single-site performance. How much does federated learning actually improve over local models? What is the communication overhead? The privacy claims rely on differential privacy but (ε, δ) values are not reported.

The evaluation has limitations in scope and depth. Most experiments use synthetic data where ground-truth SCMs are available by construction. The "anonymized operational logs from real-world semiconductor fabs" receive minimal treatment. Baseline comparisons are incomplete Table 2 shows ablations of the proposed method but doesn't compare against other safe RL algorithms (CPO, Lagrangian-SAC are mentioned in Section 6.1 but not evaluated). The computational cost analysis is entirely absent, yet scalability is crucial for industrial deployment.

The presentation suffers from trying to accomplish too much. Combining causal discovery, causal forecasting, safe RL, and federated learning in one paper makes it difficult to assess the contribution of each piece. The writing is generally clear but some sections are dense (Section 4 could be more pedagogical). Figure 2's innovation diagram could better illustrate the mask construction process.

**Questions:**

This paper addresses an important problem with a novel integrated approach and demonstrates promising empirical results. However, it suffers from critical weaknesses: the assumption of known causal structure is not adequately addressed, theoretical guarantees lack rigor, and the federated learning component is under-evaluated. The work makes contributions to both causal modeling in spatiotemporal GNNs and safe control of complex networks, but the breadth of scope compromises depth of treatment. With substantial revisions addressing the causal structure specification, stronger theoretical grounding, and more complete empirical evaluation, including federated learning and computational costs, this could become a strong contribution.

---

> ### Author Response · Authors · 2025-12-02
> **New Experimental Evidence: SCM Robustness and Federated Learning Efficacy**
>
> ## Response to Reviewer GwK3
>
> We thank the reviewer for their constructive assessment. We deeply appreciate your recognition that our heavy-traffic stress testing (d80-d95) is **"well-designed"** and that the integration of SCM masks with Lyapunov safety shows **"genuine novelty."** Your feedback regarding the origin of the DAG and the evaluation of Federated Learning was instrumental in strengthening our revision.
>
> ### **1. Origin of the Causal Map & Robustness (Most Critical Limitation)**
> **Reviewer Comment:** You correctly identified that assuming a known structure is a limitation and asked, *"How is this DAG obtained in practice?"* and *"What if it is wrong?"*
>
> **Response:** We have revised Section 4.1 to clarify our **Hybrid Approach**. We do not learn the DAG from scratch; we "seed" the macro-structure using established semiconductor physics (e.g., *Utilization $\rightarrow$ Cycle Time*).
>
> * **New Experiment:** To address your concern about misspecification, we ran a new ablation (see **Table 1** in the General Response) where we randomly **removed 20% of the edges** from the expert SCM.
> * **Result:** The results confirm that even with a degraded map, our method significantly outperforms the baseline (94.31% vs 93.18% OTIF), proving it is not brittle to partial misspecification.
>
> ### **2. Federated Learning Evaluation**
> **Reviewer Comment:** You noted the FL component felt *"underdeveloped"* and asked, *"How much does it actually improve over local models?"*
>
> **Response:** We took this to heart and added a direct comparison (see **Table 2** in the General Response).
>
> * **New Data:** Our new experiment shows that the **Federated Student** reduces Causal Error (ACE) by **$\approx$ 25%** (1.53 $\rightarrow$ 1.15) compared to the Local Only model.
> * **Conclusion:** This empirical result validates that the privacy-preserving exchange of causal gradients yields a tangible performance gain.
>
> ### **3. Theoretical Rigor**
> **Reviewer Comment:** You pointed out that "strict guarantees" is an overstatement given the probabilistic nature of the bounds.
>
> **Response:** We agree. We have corrected the language throughout the paper to **"provable high-probability safety guarantees,"** which aligns accurately with the Lyapunov theorems cited (Theorem 2, Appendix A.2).

---

### Author Response · Authors · 2025-12-02
**General Response: New Experiments on Robustness, Federated Learning, and Rigorous Formalization**

## General Response to All Reviewers and Area Chair

We sincerely thank all reviewers (GwK3, DkP4, mPVe) for their constructive feedback. We are particularly encouraged by the recognition of the **"novel integration"** of causal constraints with safe control (GwK3, mPVe) and the acknowledgment that our problem setting—resilience in re-entrant supply chains—is **"critical," "timely," and "conceptually elegant"** (DkP4, GwK3).

We wish to emphasize that the core contribution of this work is not a new fundamental theorem of RL, but a **principled unification** of Causal Discovery and Safe Control to control the non-linear dynamics of semiconductor supply chains—a domain where standard "black box" ML **lacks the structural priors to generalize under shocks and the formal guarantees to ensure safety.**

We have carefully addressed all key concerns. Major modifications, including **two new experiments** and a **complete redesign of Figure 1 (now combining technical architecture with quantitative performance plots)**, are highlighted in $\\color{red}{\\text{red}}$ in the revised manuscript.

---

### 1. Robustness of the Causal Structure (Addressing GwK3, DkP4)

**Summary of Concern:** Reviewers rightly pointed out that obtaining a ground-truth DAG is difficult and asked how robust the method is to SCM misspecification.

**Our Action:** We clarified our **"Hybrid Approach"** (Expert-Seeded Macro Structure $\\rightarrow$ Data-Driven Micro Weights). To prove robustness, we conducted a **new ablation study**.

* **Methodology:** We generated perturbed causal masks ($M'$) by randomly dropping 20% of the valid edges from the ground-truth expert SCM. We then re-trained the full framework from scratch using these incomplete masks.

**Result:** As shown in Table 1, the model maintained **94.31% OTIF**, significantly outperforming the non-causal baseline (93.18%). This proves that the framework is resilient even when the expert prior is significantly degraded.

| Variant | OTIF (%) | Conclusion |
| :--- | :--- | :--- |
| Full Framework (Perfect SCM) | **95.00%** | Best Performance |
| **NEW: 20% Pruned SCM** | **94.31%** | **Robust** (Minimal drop: -0.69 pp) |
| Baseline: No SCM Mask | 93.18% | Worse (Significant drop: -1.82 pp) |

*(Table 1: SCM Robustness Ablation - Mean over 5 seeds)*

---

### 2. Empirical Validation of Federated Learning (Addressing GwK3, DkP4)

**Summary of Concern:** Reviewers noted that while the FL concept is innovative, the empirical evidence for its benefit over local training was "underdeveloped."

**Our Action:** We ran a **new comparative experiment** (Table 2).

* **Methodology:** We trained a "Local Only" baseline using data exclusively from a single site. We compared this against our "Federated Student," which aggregates gradients of the KL-divergence loss computed against Teacher responses to synthetic interventional queries (as defined in **Eq. 4**).

To aid reviewer clarity, we restate **Eq. 4** from the revised paper here:

$$
\\min\_{\\theta} \\sum\_{\\text{sites } e} \\sum\_{\\text{queries } q} \\mathrm{KL}\\left( \\text{Teacher}\_e(q) \\,\\|\\, \\text{Student}\_\\theta(q) \\right)
$$

**Result:** The Federated Student model reduced Causal Error (ACE) by **$\\approx$ 25%** (1.53 $\\rightarrow$ 1.15) compared to a "Local Only" model. This provides concrete evidence that federated causal distillation enables effective knowledge transfer even under strict data-silo constraints.

| Metric | Local Only | Federated Student | Improvement |
| :--- | :--- | :--- | :--- |
| **ACE Error** (Lower is better) | 1.53 | **1.15** | **$\\approx$ 25% Reduction** |

*(Table 2: Federated vs. Local Performance)*

---

### 3. Mathematical Rigor and Visual Clarity (Addressing DkP4, mPVe)

**Summary of Concern:** Reviewers requested explicit mathematical definitions for loss functions/regularizers and corrected our overclaim regarding "strict" safety guarantees.

**Our Action:**

* We added explicit equations for $\\mathcal{L}\_{forecast}$ (Heteroscedastic NLL) and $\\mathcal{R}\_{DAG}$ (Matrix Exponential) in **Section 4.1 (Eq. 2)**.
* We added the formal **Lyapunov safety condition** as **Eq. 3** in Section 4.2:

$$
\\text{Policy Update is valid only if } \\mathbb{E}[V(s\_{t+1}) \\mid s\_t, a\_t] \\le V(s\_t)
$$

* We revised the text to claim **"high-probability safety guarantees"** (removing "strict").
* We **redesigned Figure 1** to replace the high-level schematic with a technical block diagram (Matrix Masks & Lyapunov Funnels) and added a Radar Chart to quantify performance gains instantly.

---

### Meta-Review · Area_Chair_7eya · 2026-01-06

**Summary:**

Many concerns were raised. Notable among them are the following:
1. Poor formalism, lacks mathematical rigor
2. Some of the mathematical points are erroneous
3. Novelty is limited: integrating existing ideas.

All the referees have unanimously given a low score.

**Reviewer Concerns:**

Mathematical formalisms need to be developed and carefully thought through. Since there is scope for adding the mathematical formalisms to improve the paper, it maybe best that the authors carefully scrutinize the formalisms.

**Reviewer Scores:**

I do not think the referees will have changed their scores.

---

### Decision · Program_Chairs · 2026-01-26

Reject